# Disorder engineering and conductivity dome in ReS$_2$ with electrolyte gating

Dmitry Ovchinnikov[1,2], Fernando Gargiulo[3], Adrien Allain[1,2], Diego José Pasquier[3], Dumitru Dumcenco[1,2], Ching-Hwa Ho[4], Oleg V. Yazyev[3] & Andras Kis[1,2]

Atomically thin rhenium disulphide (ReS$_2$) is a member of the transition metal dichalcogenide family of materials. This two-dimensional semiconductor is characterized by weak interlayer coupling and a distorted 1T structure, which leads to anisotropy in electrical and optical properties. Here we report on the electrical transport study of mono- and multilayer ReS$_2$ with polymer electrolyte gating. We find that the conductivity of monolayer ReS$_2$ is completely suppressed at high carrier densities, an unusual feature unique to monolayers, making ReS$_2$ the first example of such a material. Using dual-gated devices, we can distinguish the gate-induced doping from the electrostatic disorder induced by the polymer electrolyte itself. Theoretical calculations and a transport model indicate that the observed conductivity suppression can be explained by a combination of a narrow conduction band and Anderson localization due to electrolyte-induced disorder.

[1] Electrical Engineering Institute, École Polytechnique Fédérale de Lausanne (EPFL), CH-1015 Lausanne, Switzerland. [2] Institute of Materials Science and Engineering, École Polytechnique Fédérale de Lausanne (EPFL), CH-1015 Lausanne, Switzerland. [3] Institute of Physics, École Polytechnique Fédérale de Lausanne (EPFL), CH-1015 Lausanne, Switzerland. [4] Graduate Institute of Applied Science and Technology, National Taiwan University of Science and Technology, Taipei 106, Taiwan. Correspondence and requests for materials should be addressed to A.K. (email: andras.kis@epfl.ch).

Rhenium disulphide ($ReS_2$) is a member of the family of recently rediscovered transition metal dichalcogenides (TMDCs). In contrast to the more widely studied $MoS_2$, which preferentially crystallizes in the 2H phase[1], $ReS_2$ has a 1T′ distorted crystal structure[2–4], which results in anisotropic optical, electrical and vibrational properties[4–7]. Recent Raman spectroscopy[3,8] and photoluminescence measurements[3] indicate that atomic layers in 1T′ $ReS_2$, unlike those of $MoS_2$, are decoupled from each other[3], which could give rise to direct bandgap preservation from monolayers to bulk crystals. This makes $ReS_2$ interesting not only in the monolayer but also in the bulk form for electronic and optoelectronic applications where its optical anisotropy could in principle allow the fabrication of polarization-sensitive photodetectors[9,10]. Field effect transistors and integrated circuits made of $ReS_2$ have already been reported[7,11–15], showing anisotropic electrical behaviour and mobilities of $1–30\,cm^2\,V^{-1}\,s^{-1}$ at room temperature in a limited range of electron doping (below $10^{13}\,cm^{-2}$).

Here we use polymer electrolyte gating of mono- and multilayer $ReS_2$ to explore a wider range of doping levels in the first $ReS_2$ electrical double-layer transistor (EDLT)[16–23] reported to date. The use of polymer electrolytes can result in charge carrier densities as high as $10^{15}\,cm^{-2}$ (ref. 24), largely exceeding doping levels that can be achieved using standard solid gates. Polymer electrolytes are however a known source of disorder. Ions from the polymer electrolyte are in direct contact with the conductive channel[25] and act as charged impurities, which degrade the mobility of charge carriers[25,26]. We also include solid bottom gates in our devices, which allow us to modulate the charge density at low temperatures where the polymer electrolyte is frozen and to disentangle the effects of doping and electrolyte-induced disorder. The aim of this work is to explore the effects of doping and disorder on the electrical conductivity of $ReS_2$. At high doping levels, a complete and reversible suppression of conductivity in monolayer $ReS_2$ is observed. In multilayer flakes the effect is milder and an insulator–metal–insulator sequence is measured instead. Our band structure and transport calculations furthermore shed light on the mechanisms of conductivity suppression.

## Results

**Current suppression at high doping levels in monolayer $ReS_2$.** We first focus on monolayer $ReS_2$ EDLTs. Device schematic is shown on Fig. 1a (see Methods and Supplementary Methods for the details of device fabrication and EDLT measurements).

On Fig. 1b current $I_s$ as a function of polymer electrolyte voltage ($V_{PE}$) is presented. Strikingly, the current falls below the instrumentation noise floor at high carrier densities after reaching a maximum. Four-probe measurements reveal that sheet conductivity $G$ reproduces this behaviour, thus ruling out the possibility of a contact resistance effect (Supplementary Fig. 1 and Supplementary Note 1). In the main text we concentrate on devices measured using PS-PMMA-PS:[EMIM]-[TFSI] as the electrolyte. Experiments using $LiClO_4$-based polymer electrolyte gave essentially the same result (Supplementary Fig. 2 and Supplementary Note 2). Although a strong hysteresis is present in our measurements, the initial conductivity is restored at the end of the voltage sweep, indicating that no degradation has occurred in our device. Furthermore, we swept $V_{PE}$ 10 times in a different device (Supplementary Fig. 3 and Supplementary Note 3), and found good current stability. The distinct behaviour of the conductivity with suppression at high carrier densities was observed in all monolayer $ReS_2$ devices studied (six monolayer $ReS_2$ EDLTs).

Since the observed effect is not related to contact resistance, electrolyte type or cycling history, we consider electrolyte-induced disorder as the possible origin of the observed conductivity suppression at high doping levels. To reveal the possible influence of polymer electrolyte on the conductivity, we perform consecutive measurements on the same device before and after deposition of the electrolyte, as a function of temperature. In addition to the polymer electrolyte, we use a back-gate stack containing a high-$\kappa$ dielectric to modulate the charge density in our Hall bar devices (Fig. 2d and Supplementary Methods).

The back-gate voltage $V_{bg}$ dependence of the sheet conductivity $G$ extracted from four-probe measurements for different temperatures is shown on Fig. 2a. On the left panel, the conductivity before the electrolyte deposition is shown. In the subsequent panels, $G$ as a function of $V_{bg}$ is recorded after freezing the electrolyte at a given $V_{PE}$ (freezing point $\sim 180 - 230\,K$, Supplementary Methods). Without the electrolyte, we observe a metal–insulator transition around $V_{bg} = 5.6\,V$ and field-effect mobilities of $\mu_{FE} \sim 3\,cm^2\,V^{-1}\,s^{-1}$, consistent with other studies of $ReS_2$ (refs 7,11,14,15). As soon as the electrolyte is deposited and $V_{PE} = 0\,V$ is applied (second panel), the overall conductivity decreases and the sample displays a purely insulating behaviour. Increasing the $V_{PE}$ further results in a gradual decrease of conductivity (Fig. 2a, from left to right). To quantify the changes in insulating behaviour, we tracked the conductivity dome as a distinct feature in our experiments. We fit our data in this insulating state with the thermally activated

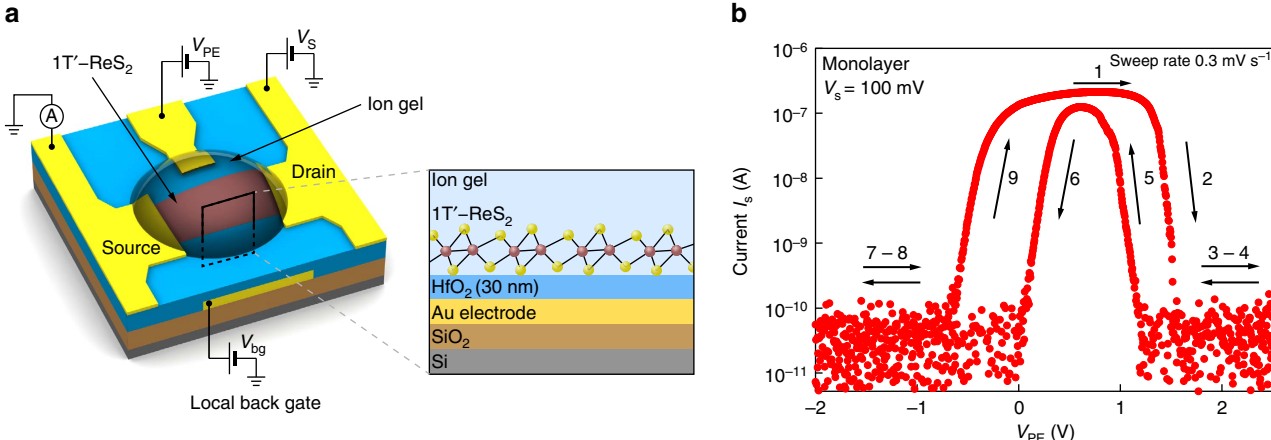

**Figure 1 | Room-temperature characterization of monolayer $ReS_2$.** (**a**) Schematic of the EDLT based on monolayer $ReS_2$. (**b**) Current $I_s$ as a function of polymer electrolyte voltage $V_{PE}$. Arrows are showing the voltage sweep direction. Conductivity is restored after full sweep.

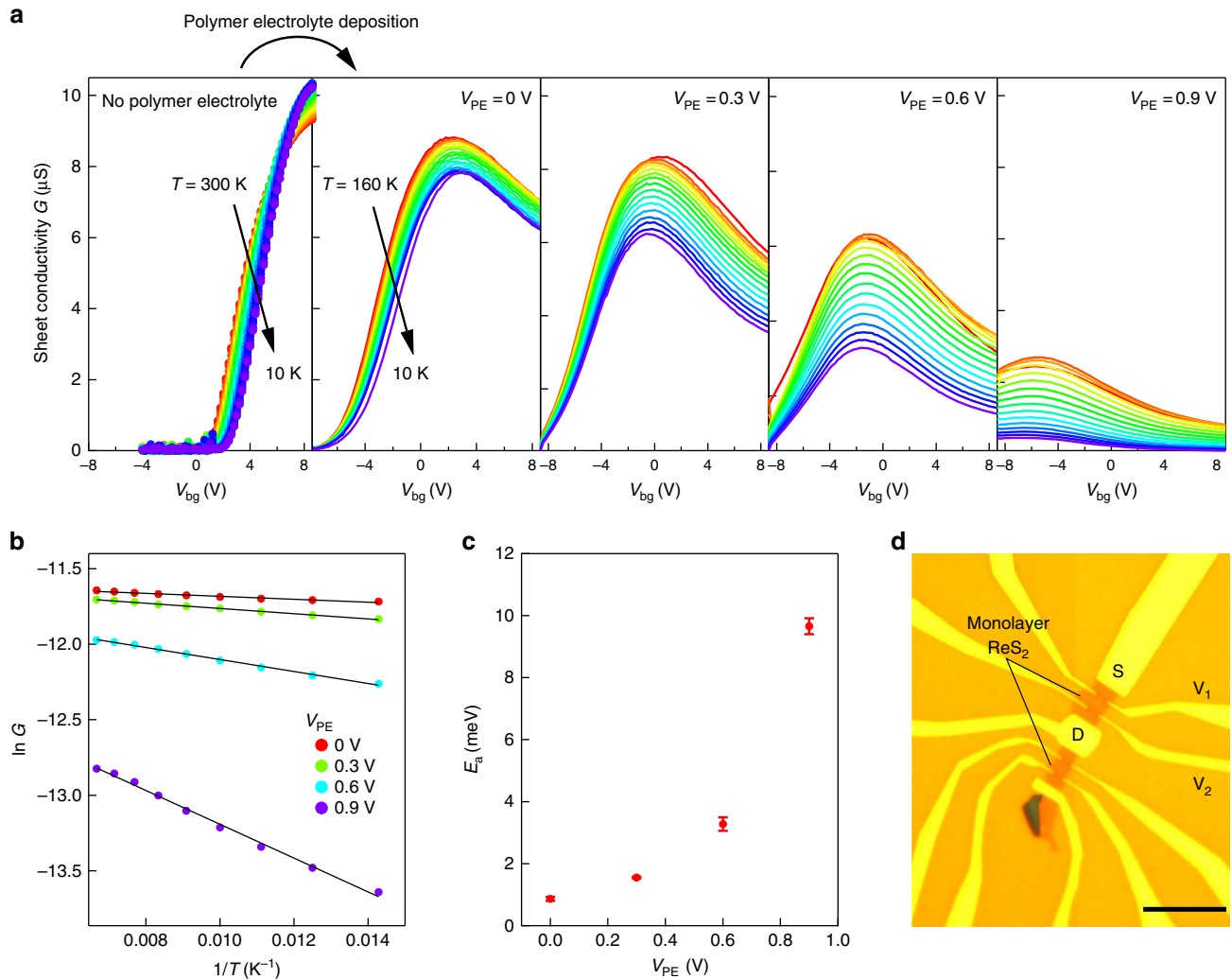

**Figure 2 | Monolayer ReS$_2$ with and without the polymer electrolyte.** (**a**) Sheet conductivity $G$ extracted from four contact measurements as a function of back-gate voltage $V_{bg}$ before PE deposition (left panel) and after PE deposition with different $V_{PE}$ applied. Colours from red to blue correspond to 300–10 (leftmost panel) and 160–10 K (the rest four panels). (**b**) Arrhenius plots for $E_a$ extracted on top of conductivity dome at different values of $V_{PE}$. Solid black lines correspond to linear fits to the equation $G(T)=G_0 e^{-E_a/k_B T}$. (**c**) Activation energy $E_a$, extracted from the top of the conductivity dome as a function of $V_{PE}$. Error bars originate from the errors in linear fit in **b**. (**d**) Optical micrograph of a monolayer ReS$_2$ multiterminal device used in this study. Scale bar, 10 μm long.

transport model $G(T)=G_0 e^{-E_a/k_B T}$, where $G_0$ is a constant conductivity, $E_a$ is the activation energy, $k_B$ the Boltzmann constant and $T$ the temperature. For all values of $V_{PE}$ we could achieve a good fit in the range between 70 and 150 K (Fig. 2b). Figure 2c shows $E_a$ as a function of $V_{PE}$ for $\Delta V_{bg}=0$ V, where $\Delta V_{bg}=V_{bg}-V_{bg}^{max}$. We can see that increasing the electrolyte voltage results in a significant increase of the activation energy (Fig. 2c). This is in contrast to band-like transport and metallic state emerging at high carrier densities in the case of solid-gated devices before electrolyte deposition. The same behaviour was observed in devices fabricated on thicker SiO$_2$ substrates (Supplementary Fig. 4 and Supplementary Note 4).

**Comparison with multilayer ReS$_2$.** Further evidence of a major role of disorder comes from our analysis of the thickness dependence. We performed similar measurements on ReS$_2$ with thicknesses ranging from 0.75 (monolayer) to 21 nm. Figure 3a presents the room-temperature field-effect curves of ReS$_2$ EDLTs with different thickness. For clarity, only sweeps in the reverse direction, from $V_{PE}=2.5$ to $-0.5$ V, are shown. There is a stark contrast between the case of monolayer and thicker layers. First,

monolayer EDLTs are the only devices that switch off at high doping levels, while the multilayers, although displaying a conductivity dome, remain largely conductive at $V_{PE}=2.5$ V. Second, multilayer devices are systematically 4–8 times more conductive than monolayers in the ON state. Among multilayers (>2 layers (L)), device-to-device variation in doping and hysteresis are of the same order of magnitude, making these curves essentially undistinguishable. Although it is expected that monolayers are more sensitive to surface disorder, such a drastic difference was not observed between monolayer and multilayer MoS$_2$, WSe$_2$ or MoSe$_2$ (Supplementary Note 5).

We have performed measurements at different temperatures using a 10 nm thick flake as a representative of multilayer ReS$_2$ (Supplementary Fig. 6 and Supplementary Note 6). On Fig. 3b we summarize our measurements on this device. Moving from the conduction band edge, the device shows weakly insulating behaviour. Around $V_{PE}=0.5$ V, transition to the metallic regime occurs, which is consistent with recent measurements on multilayer ReS$_2$ with a solid gate[12]. After the conductivity dome, the device undergoes a transition back to the insulating state around $V_{PE}=2.3$ V.

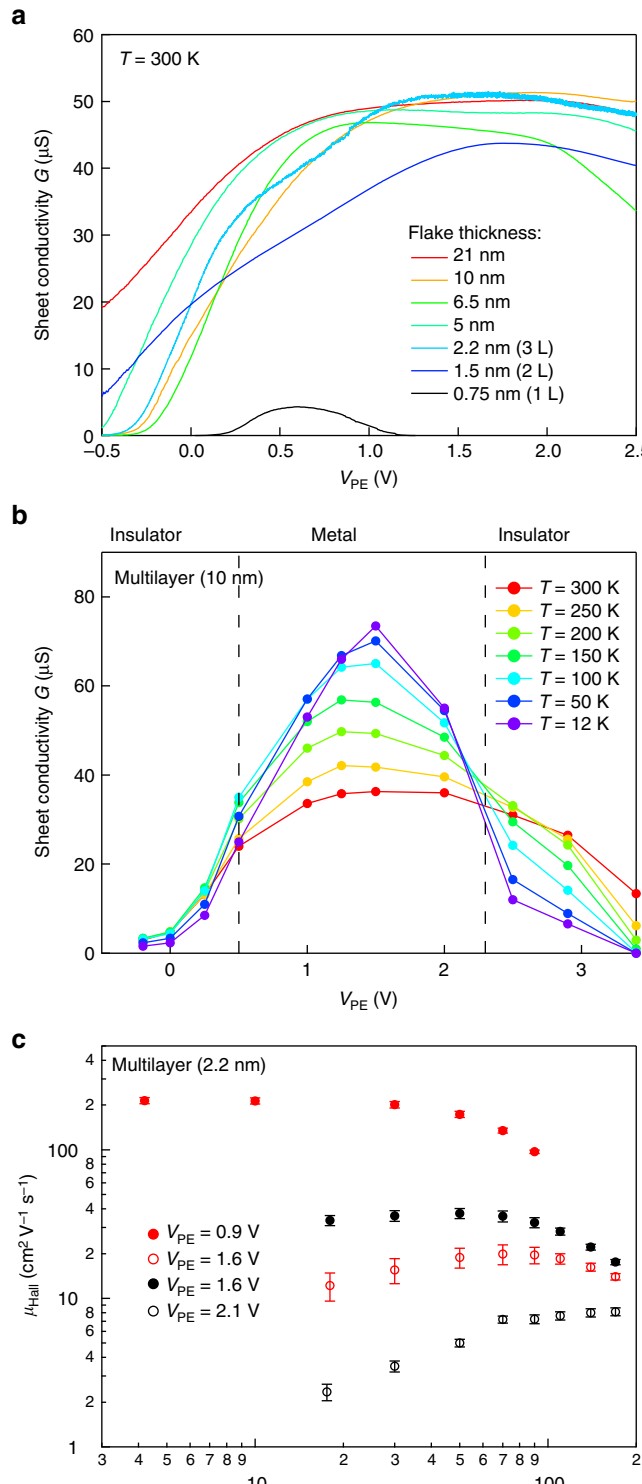

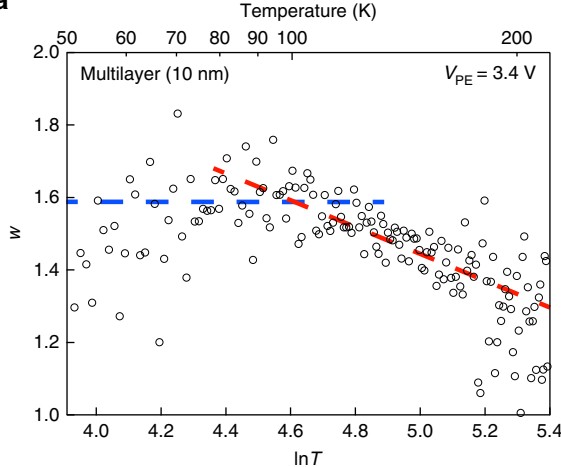

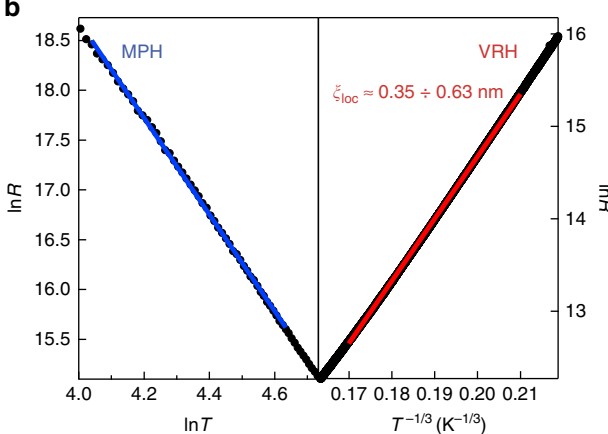

**Figure 4 | Insulating state of multilayer ReS$_2$ at high carrier densities.**
(**a**) Reduced activation energy $w$ as a function of $\ln T$. Red and blue dashed lines correspond to variable range hopping (VRH) and multiphonon hopping (MPH) regimes, respectively. (**b**) Fits for MPH (left) and VRH (right) in the corresponding range of temperatures.

**Figure 3 | Transport in multilayer ReS$_2$ with polymer electrolyte gating.**
(**a**) Sheet conductivity $G$ as a function of $V_{PE}$ for ReS$_2$ flakes of different thicknesses. (**b**) Insulator–metal–insulator sequence for multilayer (10 nm) ReS$_2$ flake. Dashed lines are pointing on the regions of $V_{PE}$, where transitions are occurring. (**c**) Hall mobility for the easy axis of trilayer ReS$_2$ (2.2 nm thick) as a function of temperature $T$ for different carrier densities and $V_{PE}$ in the metallic state. Red markers correspond to fixed carrier density $n_{2D} = 1.55 \times 10^{13}$ cm$^{-2}$; filled, $V_{PE} = 0.9$ V; empty, $V_{PE} = 1.6$ V. Black markers, fixed carrier density $n_{2D} = 1.82 \times 10^{13}$ cm$^{-2}$; filled, $V_{PE} = 1.6$ V; empty, $V_{PE} = 2.1$ V. Error bars originate from the uncertainty in carrier density extraction from Hall effect and conductivity measurements.

We first examine evolution of conductivity in the metallic state around the conductivity dome. We have performed Hall effect measurements in another multilayer (three layers, 2.2 nm thickness), where we have measured simultaneously Hall mobility $\mu_{Hall}$ and carrier density $n_{2D}$ by taking into account the anisotropy of ReS$_2$ (Supplementary Fig. 7 and Supplementary Note 7). Carrier densities of up to $2.3 \times 10^{13}$ cm$^{-2}$ could be induced at high-positive $V_{PE}$. We have performed three cooldowns: for charge densities left of the conductivity dome ($V_{PE} = 0.9$ V); near the conductivity maximum ($V_{PE} = 1.6$ V); and right above it ($V_{PE} = 2.1$ V). Cooldowns were performed at close values of $V_{PE}$ and we could continuously modulate the carrier density between neighbouring cooldowns by applying $V_{bg}$ to the silicon substrate covered by 270 nm SiO$_2$ (Supplementary Fig. 8b). This allowed us to measure the Hall mobility for each specific value of $V_{PE}$ and $n_{2D}$. The striking feature of our measurements is the significant decrease of mobility at the same values of carrier density for increasing values of $V_{PE}$, as shown on Fig. 3c. During the first cooldown ($V_{PE} = 0.9$ V), we measured metallic behaviour with Hall mobility values exceeding 200 cm$^2$ V$^{-1}$ s$^{-1}$ at low temperatures (red filled markers). With $V_{PE}$ increasing to 1.6 V, the mobility at the same carrier density decreases by a factor of 8 (red empty markers). The same behaviour is observed while moving from the second to the third

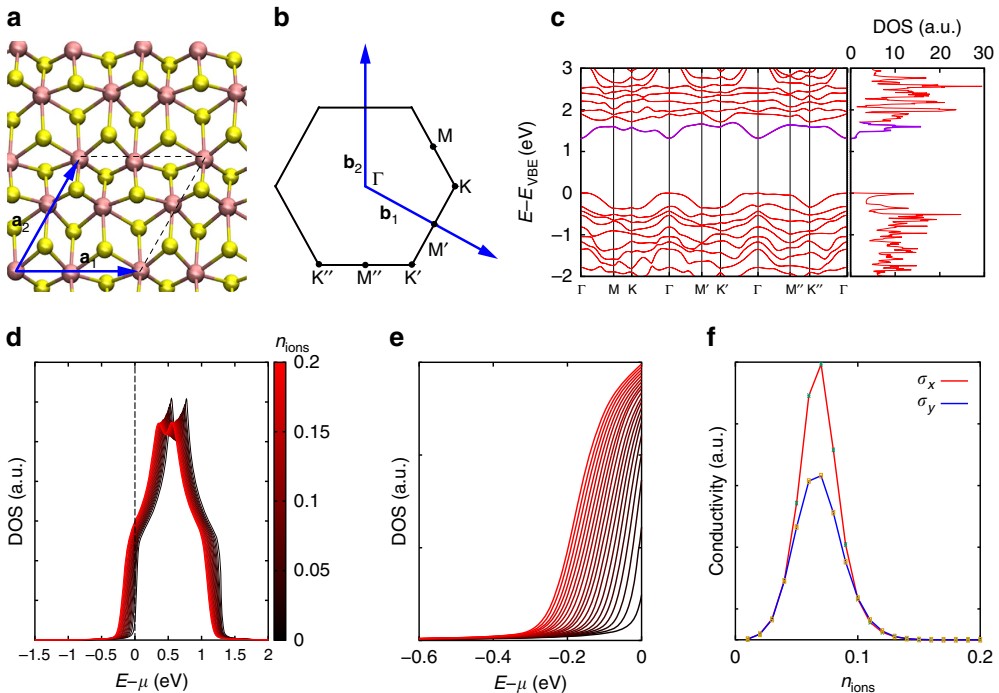

**Figure 5 | Electronic structure of monolayer ReS$_2$ from *ab initio* calculations.** (**a**) Ball-and-stick representation of the atomic structure of monolayer ReS$_2$. Lattice vectors (**a$_1$**,**a$_2$**) and unit cell (dashed lines) are illustrated. (**b**) Brillouin zone and primitive vectors (**b$_1$**,**b$_2$**) of reciprocal lattice. (**c**) Energy bands calculated along high-symmetry directions connecting the vertices defined in **b** and $k$-integrated density of states (right panel). The lowest energy conductance band, as well as its contribution in the DOS are highlighted in purple. (**d**) Density of states modification due to addition of ions with the concentration $n_{ions} = (N_+ - N_-)/N_{cells}$ on top of conductivity channel calculated using our transport model. Colour code corresponds to the amount of ions $n_{ions}$. (**e**) Close-up view of the conduction band edge. (**f**) Conductivities $\sigma_x$ and $\sigma_y$ as a function of ionic concentration $n_{ions}$ for directions parallel and perpendicular to the easy axis (along Re chains), respectively, calculated using the Kubo formula.

cooldown. Finally, the material becomes insulating as both the $V_{PE}$ and $n_{2D}$ are increased (Supplementary Fig. 8).

The insulating state at high carrier densities is a distinct feature of multilayer ReS$_2$ EDLTs, in contrast to other semiconducting TMDCs[16,18,27], which exhibit band-like transport (see discussion in Supplementary Note 5). On Fig. 4a, we show the dependence of the reduced activation energy $w = -\mathrm{d}(\ln R)/\mathrm{d}(\ln T)$ (ref. 28) on temperature for the 10 nm thick flake previously discussed on Fig. 3b. We distinguish two types of behaviour. In the $96 - 172\,\mathrm{K}$ temperature range, the temperature dependence of the resistance can be fitted using the Mott variable range hopping[29] behaviour $R \propto \exp[(T_0/T)^{1/3}]$. The coefficient extracted from the $w - \ln T$ dependence is 0.36 (red line on Fig. 4a), which fits well to the variable range hopping model. We find a density of states $D_{2D}^{theory} = 4.17 \times 10^{14}\,\mathrm{eV}^{-1}\,\mathrm{cm}^{-2}$ and localization length $\xi_{loc} \approx 0.35 \div 0.63\,\mathrm{nm}$ (see also Supplementary Note 8 for effective mass calculations). At lower temperatures, $w$ reaches saturation as a function of $\ln T$, which is the indication of a multiphonon hopping regime. We discuss both conduction mechanisms further in Supplementary Notes 9 and 10.

Activation and hopping regimes observed in ReS$_2$ at high carrier densities suggest that disorder plays an important role in the observed behaviour. Aside from it, there are other possible explanation, which should be considered: phase transition due to doping, complete filling of the disentangled conduction band (see further text and Supplementary Note 8 for discussion of the ReS$_2$ band structure) and influence of the perpendicular electric field on the band structure. These are discussed and ruled out in Supplementary Note 11.

**Theoretical modelling.** We have performed density functional theory calculations of the band structure of mono- and multilayer

ReS$_2$ to shed more light on the observed behaviour of electrical conductance. Figure 5a–c shows the crystal structure of monolayer ReS$_2$ together with the calculated band structure along high symmetry directions and the integrated density of states (DOS). We find an unusual feature in the band structure—a narrow conduction band almost separated from other bands by a minimum in the DOS, as shown in Fig. 5c, right panel. This feature is present in both mono- (Fig. 5c) and multilayer ReS$_2$ (Supplementary Fig. 9).

Further on, we concentrate on the quantitative interpretation of our findings. Depending on the effective strength of the interaction between ions and electrons (holes), and the effective mass of the charge carriers, the latter may form a bound state preventing transport. Monolayer ReS$_2$ has a narrow conduction band $\approx 0.4\,\mathrm{eV}$ wide and a large effective mass $m^\star = 0.5\,m_e$, where $m_e$ is the free-electron mass (for effective masses and bulk ReS$_2$ band structure see Supplementary Note 8). This makes ReS$_2$ similar to organic semiconductors such as $p$-doped rubrene (highest occupied molecular orbital bandwidth $D \approx 0.4\,\mathrm{eV}$, hole effective mass $m_h^\star = 0.6\,m_e$ (ref. 30)) that shows a decrease of conductivity at high charge densities[26,31–33], albeit without a full conductivity suppression at high charge densities like in the case of monolayer ReS$_2$.

A fully quantum argument based on Anderson localization sheds light on this reasoning. The ionic positions at the electric double layer are to a large extent random, thus introducing a Coulomb potential that does not reproduce the periodicity of the semiconductor lattice. Assuming that one-electron states of the conduction band of ReS$_2$ can be described by the tight-binding model, the electrolyte-induced disorder consists of a random but spatially correlated distribution of on-site energies characterized by a finite width $W$. Classical Anderson localization theory

predicts that disorder causes full localization of the one-particle states of two-dimensional (2D) lattices, irrespective of the disorder strength $W$ (refs 34,35). Nevertheless, a larger amount of disorder $W$ is generally associated with a shorter localization length $\xi$ (refs 36,37). Electronic transport in the presence of a localized spectrum takes place by means of hopping between localized states with a temperature dependence characteristic of an insulator. However, as Anderson localization is ultimately a consequence of destructive interference of the wave functions, phase-breaking mechanisms (for example, electron–phonon scattering) that take place over a phase conservation length $L_\phi \leq \xi$ prevent the physical realization of Anderson localization.

An increase of the number of ions at the electrolyte–semiconductor interface translates into the broadening of the overall on-site energy distribution, so that the effective amount of disorder $W$ increases. Therefore, $\xi$ is a decreasing function of $V_{PE}$. If one assumes that, at fixed temperature, $L_\phi$ does not vary considerably with doping, the condition for the onset of the metal–insulator transition is $\xi(V_{PE}^*) \approx L_\phi$. For increasing gate voltage ($V_{PE} > V_{PE}^*$), charge-carrier mobility $\mu(V_{PE})$ is expected to drop faster than inverse linear law, leading to a rapid decay of conductivity $\sigma \propto n\mu$. We stress that the narrowness of the conduction band is crucial to revealing the wave function localization, as the key quantity in Anderson localization is the adimensional disorder strength $W/D$ (refs 36,38). In our opinion, this phenomenon is responsible for such a peculiar behavior of monolayer ReS$_2$ among other 2D TMDCs.

To investigate qualitatively the discussed phenomenon, we consider the following model of electronic transport. We describe the lowest conduction band of ReS$_2$, highlighted in Fig. 5c, with a tight-binding model on a rectangular lattice, with $x$ and $y$ directions corresponding to the parallel and perpendicular directions relative to the easy axis (along Re chains). Monovalent point charges (for example, Li$^+$) are placed at a distance $\Delta z = 20$ Å from the plane of the rectangular lattice. To guarantee the total charge neutrality of the ionic gate-semiconductor interface, the ionic concentration $n_{ions}$ must be equal to the electron concentration $n_{2D}$. The electrical conductivity has been calculated by means of the Kubo formula[39,40] assuming linear response to the applied electric field. Further details could be found in Supplementary Note 12.

The calculated DOS shown in Fig. 5d is characterized by long tails on increasing $n_{ions}$. These tails indicate the presence of localized states induced by electrostatic disorder. Moreover, $n_{ions}$ determines the chemical potential $\mu$ of the electrons, which shifts further into the conduction band on the increase of doping. The transport behaviour as a function of $n_{ions}$ is ultimately determined by the interplay between the increasingly localized states of the spectrum and the position of the chemical potential within the conduction band. The conductivity $\sigma$ calculated along $x$ and $y$ directions is shown in Fig. 5e. Here a pronounced dome in the conductivity followed by its full suppression at high carrier densities is observed. The ionic concentration, that is, the carrier density associated with the peak of the dome, is $n_{ions}^* = 0.06 \div 0.08$ ions per unit cell. We observe a very good agreement with measured carrier densities extracted from the Hall effect data (Supplementary Fig. 8a). The anisotropy calculated in the region of the dome, $\sigma_y/\sigma_x = 0.6$ also agrees well with our experimental data. Above certain ionic concentrations the curves along the two directions merge and become undistinguishable. This isotropic regime at high carrier densities could be clearly seen in the experimentally measured two-probe conductivity curves (Supplementary Fig. 10). We ascribe this feature to the onset of full localization in the states in the energy region around the chemical potential, that is, those states responsible for transport.

Therefore, localization eliminates any preferential direction for transport.

Theoretical intuition suggests how the behaviour of conductivity must change in multilayer ReS$_2$. First, classical scaling theory of Anderson localization in $d = 2 + \varepsilon$ ($\varepsilon > 0$) dimensions predicts that extended states do not disappear entirely, but in the energy spectrum they are separated from localized states by so-called mobility edges. Second, the injection of electrons itself into the conduction band of ReS$_2$ results in a rapid screening of the Coulomb potential in the bulk of the sample, which is less affected by electrostatic disorder (Supplementary Fig. 11), thus preserving the charge-carrier mobility. These arguments point towards a scenario where the multilayer ReS$_2$ conductivity is less influenced by disorder than in the case of the monolayer.

## Discussion

In conclusion, we have realized the first transport study of ReS$_2$ EDLT with thicknesses ranging from 1 (0.75 nm) to $\sim$30 layers (21 nm). We demonstrate that ionic disorder leads to an unusual OFF state at high carrier densities in the case of monolayers. In the case of multilayers, an insulator–metal–insulator sequence, as well as a quenching of Hall mobility with increasing $V_{PE}$ were observed. The highly doped state of multilayer ReS$_2$ is characterized by a hopping mechanisms with small localization length. Owing to the unique band structure with a narrow low-energy conduction band ReS$_2$ stands apart from other TMDCs, where such modulation of conductivity at high carrier densities was not observed. Our transport model quantitatively explains our findings.

## Methods

**Device fabrication.** Flakes of mono- and multilayer ReS$_2$ were obtained from bulk crystals, which were cleaved using an adhesive tape and transferred onto a degenerately doped n$^{++}$ Si chip covered by 270 nm SiO$_2$. Contacts were fabricated using standard e-beam lithography, followed by evaporation of Pd/Au contacts and liftoff in acetone. Selected devices were also patterned with a second e-beam step and subsequently etched in O$_2$/SF$_6$ plasma. Another series of devices was fabricated by transferring monolayer flakes on top of local back gates (Cr/Au) covered with 30 nm HfO$_2$ deposited using atomic layer deposition.

**Data availability.** The data that support the findings of this study are available from the corresponding author on request.

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

## Acknowledgements

We gratefully acknowledge the help and supervision of Prof. Y.-S. Huang with crystal growth. We thank O. Lopez Sanchez, Y.-C. Kung, K. Marinov, S. Misra and M. Audiffred for help and motivating discussions. We acknowledge the help of D. Alexander, S. Lopatin and S. Lazar for training and support with electron microscopy, which was performed using a Cs-corrected TEM (FEI Titan Themis) at the EPFL Interdisciplinary Center for Electron Microscopy (CIME). Device fabrication was carried out in the EPFL Center for Micro/Nanotechnology (CMI). We thank Z. Benes (CMI) for technical support with e-beam lithography. This work was financially supported by funding from the European Union's Seventh Framework Programme FP7/2007–2013 under Grant Agreement No. 318804 (SNM) and Swiss SNF Sinergia Grant No. 147607. The work was carried out in frames of the Marie Curie ITN network 'MoWSeS' (Grant No. 317451). We acknowledge funding by the EC under the Graphene Flagship (Grant Agreement No. 604391).

## Author contributions

D.O. conceived the experiment, fabricated devices, performed transport measurements and analysed the data; A.A. performed transport measurements and analysed the data; D.D. and C.-H.H. synthesized crystals of $ReS_2$; F.G. performed transport calculations; D.J.P. performed DOS calculations under the supervision of O.V.Y.; D.O., F.G., A.A., A.K. and O.V.Y. co-wrote the manuscript with critical input from all authors; A.K. supervised the project.

## Additional information

**Competing financial interests:** The authors declare no competing financial interests.

