## [Peer review file · Nature Communications]

Reviewers' comments:

Reviewer #1 (Remarks to the Author):

In this manuscript, the authors report that the conductivity of the monolayer ReS₂ is completely suppressed at high carrier density and conductivity dome with insulator-metal-insulator sequence in multilayer flakes. The conductivity suppress in monolayer can be explained as the combination of narrow conduction band and Anderson localization caused by electrolyte-induced disorder. The results are interesting. Therefore, the paper could be published in Nature Communications. However, the authors need to address the following points and the comments mentioned above.

(1) The authors claimed that the polymer electrolytes are the source of the disorder which caused the conductivity suppression at high carrier densities ReS₂. As we know, liquid ions gate often react with the samples which is the origin of the defects. This process is irreversible. However, the current I_s don't showing much degradation after 10 cycles of sweeping VPE. Did authors find any degradation of I_s or the damage of the samples after more cycles of sweeping VPE and what's the temperature during sweeping VPE ?

(2) The authors reported that the other monolayer TMDCs don't showing the completely suppressing of conductivity at high carrier densities. What's the main reason of the difference? Did the crystal structures of those other monolayer TMDCs are the same with the ReS₂?

(3) The theory calculation result of the narrow low-energy conduction band of ReS₂ stand apart from other TMDCs is crucial to conductivity modulation in ReS₂. Dose the conductivity can completely suppressed in monolayer ReS₂ without the localization at disorder free or reduce the disorder strength by born nitride protecting the interface heterostructure?

Reviewer #2 (Remarks to the Author):

The authors have done a thorough experimental and theoretical work to demonstrate, and understand, the suppression of conductivity arising in ReS₂ at high carrier densities due to electrolyte-induced disorder. This is a novel work and will be of interest to the scientific community working in the field of 2D materials. The overall data analyses and conclusions are sound and the manuscript can be published in Nature Communications, however, there are a few questions which the authors must address:

1. Is the sheet conductivity plotted in Figure 2(a) 2-point or 4-point? That should be explicitly specified.

2. In Figure 3(b), the authors show an insulator-metal-insulator (I-M-I) transition in a 10 nm thick ReS₂ flake, but, in order to further analyze the second transition (metal-insulator transition occurring after the conductivity dome), the authors perform Hall effect measurements on a 2.2 nm thick ReS₂ flake instead. Although the authors state earlier that among multilayer ReS₂ (> 2L) flakes, the curves are essentially indistinguishable in terms of variation in doping and hysteresis, it would be better and more consistent if the authors can show the plot for insulator-metal-insulator transition occurring in a 2.2 nm flake instead of the thicker 10 nm flake in Figure 3(b). Or conversely, the authors should show the Hall effect mobility measurements on a 10 nm thick flake instead of a trilayer flake in Figure 3(c).

The values of V_{pe} at which the I-M and M-I transitions occur will obviously be different for the 10 nm thick flake as compared to the 2.2 nm thick trilayer flake. Therefore, it is not straightforward to simply analyze the M-I transition occurring in a multilayer 10 nm thick flake (at $V_{pe} \sim 2.3$ V) by performing Hall measurements on a multilayer 2.2 nm thick flakes. It will be a lot easier to

correlate the data and conduction regimes presented in Figure 3(b) to the mobilities shown in Figure 3(c) if they are extracted from the same flake with the same thickness.

Moreover, in Figure 3(c), the Hall mobility values for $n_{2D} = 1.55 \times 10^{13}$ and $V_{pe} = 0.9$ V are not presented beyond 90 K (it is not there even in supplement figure S8). What is the reason for this?

3. In SI section 8, in lines 5-6 of the first paragraph, the authors state that they 'performed measurements in two-terminal "directional" monolayer structures (see previous section in SI)'. Where is this "previous" section the authors are pointing to? Or are they referring to the plot presented in section 12 figure S10(a)?

Reviewer #3 (Remarks to the Author):

In this manuscript, authors find disorder induced Anderson localisation which induces metal-insulator transition at high doping level as increasing the polymer electrolyte voltage. Such Anderson localisation even completely suppresses the conductivity at high doping level in monolayer ReS₂. Overall the findings are original and interesting. I would recommend to publish it after revision considering following comments:

(1) Since authors ascribe the observed completely suppression of conductivity in monolayer ReS₂ to disorder induced Anderson localisation, the signature of phase transition from insulator to metal as increasing temperature is expected.

(2) The effects of inter-layer interaction may play important roles, although such interaction is weak in ReS₂.

(3) Is the sheet conductivity of 10 nm thick ReS₂ at 300K presented in Fig. 3a and 3b not same?

(4) How large is the V_{bg} in Figure 1?

(5) The observed sheet conductivity in the ON state of multilayer devices is 4-8 times larger than that of monolayer devices. However, the difference of sheet conductivity among multilayer devices is rather small. What is the reason responsible for such large difference?

We thank the reviewers for careful reading of the manuscript, positive response and useful comments. Below, the questions are addressed point by point.

Reviewer #1 (Remarks to the Author):

In this manuscript, the authors report that the conductivity of the monolayer ReS₂ is completely suppressed at high carrier density and conductivity dome with insulator-metal-insulator sequence in multilayer flakes. The conductivity suppress in monolayer can be explained as the combination of narrow conduction band and Anderson localization caused by electrolyte-induced disorder. The results are interesting. Therefore, the paper could be published in Nature Communications. However, the authors need to address the following points and the comments mentioned above.

(1) The authors claimed that the polymer electrolytes are the source of the disorder which caused the conductivity suppression at high carrier densities ReS₂. As we know, liquid ions gate often react with the samples which is the origin of the defects. This process is irreversible. However, the current Is don't showing much degradation after 10 cycles of sweeping VPE.

Did authors find any degradation of Is or the damage of the samples after more cycles of sweeping VPE

Thank you for the questions. None of our devices showed degradation over time within months of measurements and multiple sweeps. They definitely survive more than 10 cycles and we did not find the point where degradation happens. Leakage currents were monitored to make sure nothing unusual is happening, as shown in Supplementary Figure S1. We write in the Supplementary Information, Section 2:

"On Supplementary Figure S1(b) we present the leakage current, recorded simultaneously with the drain current. The values, not exceeding 700 pA, indicate the absence of electrochemical reactions and flake degradation."

and what's the temperature during sweeping VPE ?

These measurements have been performed at $T = 300\text{K}$, we now write in the section 4 of SI:

"To rule out degradation during cycling or detrimental influence of ions on our device performance we performed cycling of V_{PE} 10 times at elevated sweep speed (50 mV/s) in the monolayer ReS₂ EDLT, the measurements were performed at 300K."

(2) The authors reported that the other monolayer TMDCs don't showing the completely suppressing of conductivity at high carrier densities. What's the main reason of the difference? Did the crystal structures of those other monolayer TMDCs are the same with the ReS₂?

ReS₂ stands out from other semiconducting TMDs, which normally have 2H (more rarely 3R) crystal structure (Mo(W)Se(S) materials). ReS₂ has stable 1T' structure, which in turn leads to the unusual band structure. We write in the text:

"We stress that the narrowness of the conduction band is crucial to revealing the wavefunction localization, as the key quantity in Anderson localization is the adimensional disorder strength W/D ^{36,38}. In our opinion, this phenomenon is responsible for such a peculiar behavior of monolayer ReS₂ among other 2D TMDCs."

(3) The theory calculation result of the narrow low-energy conduction band of ReS₂ stand apart from other TMDCs is crucial to conductivity modulation in ReS₂. Dose the conductivity can completely suppressed in monolayer ReS₂ without the localization at disorder free or reduce the disorder strength by born nitride protecting the interface heterostructure?

Thank you for the question. The localization occurs because of band structure of ReS₂ and electrostatic disorder (effect of electrostatic disorder is more strong in this material due to exotic band structure with narrow low energy CB). Protecting monolayer ReS₂ with monolayer BN will have two effects: (i) decrease of effective capacitance due to increase of distance between conductive channel and ion gel; (ii) smoothing of electrostatic disorder coming from electrolyte due to an increased distance between the conductive channel and the interface with the ionic liquid. In fact, we can compare it to the having a bilayer ReS₂ instead of monolayer, as soon as the bottom layer of the bilayer will be already screened from disorder by the topmost layer. Protection of monolayer ReS₂ by h-BN would be an interesting subject for future study, but this is beyond the scope of the presented work.

Reviewer #2 (Remarks to the Author):

The authors have done a thorough experimental and theoretical work to demonstrate, and understand, the suppression of conductivity arising in ReS₂ at high carrier densities due to electrolyte-induced disorder. This is a novel work and will be of interest to the scientific community working in the field of 2D materials. The overall data analyses and conclusions are sound and the manuscript can be published in Nature Communications, however, there are a few questions which the authors must address:

1. Is the sheet conductivity plotted in Figure 2(a) 2-point or 4-point? That should be explicitly specified.

This is sheet conductivity extracted from four-contact measurements. We now clarify it in the text:

"The back-gate voltage V_{bg} dependence of the sheet conductivity G extracted from four-probe measurements for different temperatures is shown on Figure 2(a)."

Caption of Figure 2 (a) has also been modified accordingly.

2. In Figure 3(b), the authors show an insulator-metal-insulator (I-M-I) transition in a 10 nm thick ReS₂ flake, but, in order to further analyze the second transition (metal-insulator transition occurring after the conductivity dome), the authors perform Hall effect measurements on a 2.2 nm thick ReS₂ flake instead. Although the authors state earlier that among multilayer ReS₂ (> 2L) flakes, the curves are essentially indistinguishable in terms of variation in doping and hysteresis, it would be better and more consistent if the authors can show the plot for insulator-metal-insulator transition occurring in a 2.2 nm flake instead of the thicker 10 nm flake in Figure 3(b). Or conversely, the authors should show the Hall effect mobility measurements on a 10 nm thick flake instead of a trilayer flake in Figure 3(c).

The values of V_{pe} at which the I-M and M-I transitions occur will obviously be different for the 10 nm thick flake as compared to the 2.2 nm thick trilayer flake. Therefore, it is not straightforward

to simply analyze the M-I transition occurring in a multilayer 10 nm thick flake (at $V_{pe} \sim 2.3$ V) by performing Hall measurements on a multilayer 2.2 nm thick flakes. It will be a lot easier to correlate the data and conduction regimes presented in Figure 3(b) to the mobilities shown in Figure 3(c) if they are extracted from the same flake with the same thickness.

Thank you for the comment. With figure 3b, we are simply aiming to show that for multilayers the usual I-M transition is followed by an M-I transition at higher charge densities and in contrast with behavior shown on figure 2a where the deposition of the polymer electrolyte immediately suppresses the first M-I transition. The motivation for Hall measurements shown on Figure 3c is not to identify the charge densities where the transition occurs but to show that for a certain value of a doping level, the mobility is lower for higher values of V_{pe} , i.e. higher concentrations of ions on the surface of ReS_2 . - Figure 3(b) shows an overview of the I - M - I sequence for a typical multilayer flake. In other multilayers, the situation (within measured range of thicknesses with sample to sample variation caused by such factors as small V_{th} variation) will be the same. The insulating state **will appear** at high V_{PE} in multilayers of ReS_2 in the accessible range of gate voltages with the electrolyte we use. We are certain about this fact, because of (i) highly consistent room-temperature operation with the dome-like shape and decrease of conductivity at high V_{PE} ; (ii) consistent insulating state, measured at high V_{PE} in at least two more samples (apart from the one discussed in the main text).

We can furthermore address this comment by comparing carrier density values. Both samples (10 nm and 2.2 nm flakes) are unavailable at the moment, but we can comment on the data we have. We can estimate the carrier density for the 10 nm thick sample at $V_{PE} = 0.25$ V from back-gating curves below the freezing point of the ionic liquid. $V_{th} = -28$ V \div -40 V (depending on the extraction method) and 270 nm SiO_2 dielectric results in $n_{2D} = 2.2 \div 3.2 \times 10^{12} \text{ cm}^{-2}$. This fits very well into the Hall effect data from 2.2 nm thick flake (Supplementary Figure S8(a)). In addition, no significant capacitance variation of the ion gel capacitance with flake thickness (among multilayers) is expected. All these considerations strongly point towards a **universal behavior of n_{2D} vs V_{PE}** with a small device-to-device variation due a shift in V_{th} .

As mentioned before, Figure 3(c) serves a different purpose. It shows how we **deplete mobility** in the metallic state with controllable introduction of disorder with electrolyte. Furthermore, disorder **gradually** pushes the material into the insulating state. We stress that this M-I transition **is not a sharp event** with respect to V_{PE} . The interplay and competition between increasing disorder and movement of Fermi-level further into conduction band, after which disorder eventually "wins", leads to the progression of the material into the insulating state and further strong localization effects. We corrected the main text to be more clear on this point:

"We first examine evolution of conductivity in the metallic state around the conductivity dome. We have performed Hall effect measurements in another multilayer ... "

Moreover, in Figure 3(c), the Hall mobility values for $n_{2D} = 1.55 \times 10^{13}$ and $V_{pe} = 0.9$ V are not presented beyond 90 K (it is not there even in supplement figure S8). What is the reason for this?

Unfortunately we do not have this data due to the fact that this device failed. However, we do not expect unusual behavior of mobility in the region from 90K to 180K. Our data in the

metallic state at moderate carrier densities and low V_{PE} (red filled markers on Figure 3(c)) is in full agreement with measurements from a device with a solid gate done by Pradhan et al. (Nano Lett., 2015, 15 (12), pp 8377–8384), where a typical decrease of mobility with increasing temperature in the metallic state is observed due to electron-phonon scattering. Moreover, the values of the mobility are in good agreement with those of Pradhan et al. We believe that the absence of these measurements does not influence any of the major conclusions of the manuscript.

3. In SI section 8, in lines 5-6 of the first paragraph, the authors state that they 'performed measurements in two-terminal "directional" monolayer structures (see previous section in SI)'. Where is this "previous" section the authors are pointing to? Or are they referring to the plot presented in section 12 figure S10(a)?

Thank you, it has been corrected:

"We have performed measurements in two-terminal "directional" monolayer structures (see Section 12 of SI and Supplementary Figure S10)."

Reviewer #3 (Remarks to the Author):

In this manuscript, authors find disorder induced Anderson localisation which induces metal-insulator transition at high doping level as increasing the polymer electrolyte voltage. Such Anderson localisation even completely suppresses the conductivity at high doping level in monolayer ReS₂. Overall the findings are original and interesting. I would recommend to publish it after revision considering following comments:

(1) Since authors ascribe the observed completely suppression of conductivity in monolayer ReS₂ to disorder induced Anderson localisation, the signature of phase transition from insulator to metal as increasing temperature is expected.

Thank you for the question. That's exactly what happens in case of monolayer ReS₂ and is shown on Figure 2(a). Without PE deposition we observe insulator to metal transition. While with PE deposition the metallic state disappears. We write in the main text:

"Without the electrolyte, we observe a metal-insulator transition around $V_{bg} = 5.6$ V and field-effect mobilities of $\mu_{FE} \sim 3$ cm²/Vs, consistent with other studies of ReS₂.^{7,11,14,15} As soon as the electrolyte is deposited and $V_{PE} = 0$ V is applied (second panel), the overall conductivity decreases and the sample displays a purely insulating behavior. Increasing the V_{PE} further results in a gradual decrease of conductivity (Figure 2(a), from left to right)."

(2) The effects of inter-layer interaction may play important roles, although such interaction is weak in ReS₂.

Thank you for the question. In experiments of Tongay et al (*Nat Commun* 2014, 5) photoluminescence measurements show weak interlayer interactions. We mention this in the main text:

"Recent Raman spectroscopy^{3,8} and photoluminescence measurements³ indicate that atomic layers in 1T' ReS₂, unlike those of MoS₂, are decoupled from each other,³ which gives rise to direct band gap preservation from monolayers to bulk crystals."

Layer coupling (as well as atoms presented in the lattice, crystal structure, major defects and other factors) in fact determine the band structure and specifically narrow conduction band, which is the key feature for extreme sensitivity of this material to electrostatic disorder.

(3) Is the sheet conductivity of 10 nm thick ReS₂ at 300K presented in Fig. 3a and 3b not same?

The curves of Figures 3(a) and 3(b) at 300K do not perfectly reproduce each other due to different V_{PE} sweep rates. The difference is the following - for Figure 3(a) the sweep rate was set to 0.25 mV/s. For Figure 3(b), the sweep rate between the points was the same, at each point the current was set to stabilize, furthermore, cooldown and heatup were performed for each point. After heatup, the current was again set to stabilize. In this way, the effective sweep rate is much lower for Figure 3(b). We are however sure that with this measurement schedule, we get a homogeneous doping across the sample and drift-free conductivity. We also do not exclude a small V_{th} drift due to the slow nature of such measurements. However, qualitatively, the curves look similar with a pronounced dome-like shape, despite the peak conductivity variation from 51 μ S to 36 μ S.

(4) How large is the V_{bg} in Figure 1?

Thank you for the question. We have two types of samples, discussed in the manuscript - on local back gates and on SiO₂ substrates with a global back gate. We disconnect the back gate for the devices fabricated on 270 nm SiO₂ with global back gate and leave it floating until the electrolyte is frozen due to the way we prepare substrates. Please, find the detailed explanation below:

The e-beam alignment markers and fiducial markers for localizing the flakes on the sample are embedded into the SiO₂ substrate (in order to resist adhesive-tape exfoliation) with reactive ion etching and are thus connected to the back gate. We design our devices in a way that the electrodes/flakes do not touch the markers to avoid leakage. The polymer electrolyte is spin coated over the entire surface of the chip. Because of this, if we applied 0V to the global back gate we would effectively apply 0V to the polymer electrolyte over the large area through multiple markers. If we now also applied voltage to the polymer electrolyte (PE) electrode, there would be a competition between the two voltages, which would decrease the effectiveness of gating and would prevent us from carefully monitoring the leakage current from PE electrode (since a potential difference between the PE electrode and the markers would result in an additional leakage current). The sample on Figure 1(b) therefore has a disconnected back gate.

We note however, that we do not use room temperature measurements with PE to extract mobility values, which could be influenced by a small capacitive coupling with the back gate. All the mobility values in the manuscript are extracted with electrolyte frozen. To prove that the back-gate capacitance does not change below freezing point of the ionic liquid, we extract

the value of the geometric capacitance of 270 nm SiO₂ from the Hall effect data (please refer to Supplementary Figure S8(b)). We mention this in the Supplementary Information, section 8:

"The extracted back gate capacitance for each cooldown corresponds very well to the geometric capacitance, showing that our measurements are correct."

For the devices, fabricated on local gates and thin high-k oxide substrates, where the markers do not penetrate through the oxide and thus no shunt between the local gate and electrolyte occurs (the device discussed in Figure 2) floating or grounding the bottom gate does not make any difference.

(5) The observed sheet conductivity in the ON state of multilayer devices is 4-8 times larger than that of monolayer devices. However, the difference of sheet conductivity among multilayer devices is rather small. What is the reason responsible for such large difference?

We consider the following simple model. In the bilayer the top layer is still exposed to disorder, while the bottom layer is protected from disorder, has higher mobility and provides (from parallel conduction considerations) a significant increase of conductivity, having also high carrier density (for sake of simplicity, we neglect carrier hopping between the layers). By adding more layers, we increase the number of conductive channels. At the same time, the high doping is maintained in the 1-2 topmost layers, as shown in *Yuan et al., Nat Phys 2013, 9 (9), 563–569*. Thus the channels lying further away from the gate have negligible contribution to the overall conductivity. Consequently, there is a step-like change of conductivity from monolayer to anything thicker (within the thicknesses range which we address in current work).

REVIEWERS' COMMENTS:

Reviewer #1 (Remarks to the Author):

My concerns are addressed. And I recommend this work to be published in NC.

Reviewer #2 (Remarks to the Author):

The authors have done a good job responding to all the questions and comments posed to them. Overall, I am happy with the revised manuscript and recommend publication.

Reviewer #3 (Remarks to the Author):

Authors addressed well my comments. I suggest to publish the manuscript in current format.

Dear Reviewers,

We thank you very much for your positive and constructive comments on our manuscript and for accepting it as is.

Reviewer #1 (Remarks to the Author): My concerns are addressed. And I recommend this work to be published in NC.

Thank you very much.

Reviewer #2 (Remarks to the Author): The authors have done a good job responding to all the questions and comments posed to them. Overall, I am happy with the revised manuscript and recommend publication.

Thank you very much.

Reviewer #3 (Remarks to the Author): Authors addressed well my comments. I suggest to publish the manuscript in current format.

Thank you very much.